# Anthropogenic Impacts on Bark and Ambrosia Beetle Assemblages in Tropical Montane Forest in Northern Borneo

**DOI:** 10.3390/insects16020121

**Published:** 2025-01-26

**Authors:** Evahtira Gunggot, Roger A. Beaver, Jonathan Jimmey Lucas, Sandra Geogina George, Anastasia Rasiah, Wilson V. C. Wong, Maria Lourdes T. Lardizabal, Naoto Kamata

**Affiliations:** 1Faculty of Tropical Forestry, Universiti Malaysia Sabah, Jalan UMS, Kota Kinabalu 88400, Malaysiaw.wilson@ums.edu.my (W.V.C.W.); mlourdes@ums.edu.my (M.L.T.L.); 2161/2 Mu 5, Soi Wat Pranon, T. Donkaew, A. Maerim, Chiangmai 50180, Thailand; rogerbeaver6@gmail.com; 3The University of Tokyo Chiba Forest, The University of Tokyo Forests, Graduate School of Agricultural and Life Sciences, The University of Tokyo, 770 Amatsu, Kamogawa 299-5503, Chiba, Japan

**Keywords:** species richness, species diversity, dominance, indicator species, rarefaction and extrapolation analyses, Chao-1 estimator, stochasticity, tropical rainforest, Scolytinae, Platypodinae

## Abstract

This study investigates how bark and ambrosia beetles, which bore into the bark and wood of trees and logs, vary in their species composition and distribution across three types of forests in southern Sabah, Malaysia: primary forest, disturbed forest, and rubber plantation. Using ethanol-baited traps, researchers collected data biweekly over three years. The findings revealed a rich diversity of the beetles in all types of forest, yet the species composition was highly unpredictable between forest types. The study demonstrated that anthropogenic activities, such as the indigenous use of forests for fuel and the conversion of forest into rubber plantation, significantly affect the distribution and abundance of these beetles. These changes have crucial implications for forest health and ecosystem stability. By understanding how different forest types and anthropogenic activities impact these insects, this research provides valuable insights for managing forests and protecting biodiversity in Malaysia’s tropical ecosystems.

## 1. Introduction

Tropical rainforests are renowned for their extraordinary biodiversity, hosting a vast array of species and complex ecosystem [1,2]. The order Coleoptera is widespread within tropical rainforest ecosystems, occupying a diverse array of niches and fulfilling a myriad of ecological roles [3,4]. Among the beetle group, bark and ambrosia beetles within the subfamilies Scolytinae and Platypodinae are particularly well-adapted to the tropical rainforest environment, forming intricate relationships with the dominant woody plants [5], which are ecologically diverse and economically important. The subfamily Scolytinae includes approximately 6100 species, which includes phloem feeders (bark beetles (*sensu stricto*)), fungal feeders (ambrosia beetles), and species that feed on pith, wood, fruit, seeds, and others [6]. In contrast, the subfamily Platypodinae comprises approximately 1500 species, almost all of which are ambrosia beetles [7]. Ambrosia beetles bore into the wood of trees and cultivate specific fungi within their galleries. These fungi serve as the primary food source for both adult beetles and their larvae [3].

Bark and ambrosia beetles play a critical role in forest ecosystems, often causing tree mortality, biodiversity loss, and economic damage to the forestry industry. Bark beetles (*sensu stricto*), such as the mountain pine beetle (*Dendroctonus ponderosae* Hopkins and the spruce beetle (*Ips typographus* (Linnaeus)), exhibit host specificity, with the first species predominantly infesting pine species and the latter targeting spruce trees [3]. These beetles penetrate the bark to consume the phloem, thereby disrupting the tree’s nutrient transport system, which can result in tree mortality. Outbreaks of these beetles can become severe, posing a substantial threat to forest management and economics, especially in regions where they cause high tree mortality [8]. Even when infested trees survive, the beetles’ activity can diminish the wood’s commercial value due to the creation of galleries and discoloration by associated fungi [9,10]. Over the past several decades, there has been an alarming increase in the incidence of ambrosia beetles infesting and killing seemingly healthy trees [11]. This trend has led to several notable mass mortalities of trees across different regions. In Japan and Korea, oak wilt diseases have emerged as great threats to forest ecosystems [12,13]. The southeastern USA has been grappling with the spread of laurel wilt (*Harringtonia lauricola*), which has had devastating effects on native tree species belonging to the family Lauraceae and to avocado plantations [14,15]. The USA and other countries have witnessed the emergence of wilting diseases that is carried by an ambrosia beetle, *Euwallacea fornicatus* (Eichhoff) complex and have impacted various tree populations [16,17] and have caused substantial losses in avocado plantations in the USA [18]. Southeast Asia has also experienced its share of wilting diseases vectored by non-native platypodid ambrosia beetle species affecting local flora [19]. These cases collectively highlight the growing threat posed by ambrosia beetle-fungus symbioses to both forest ecosystems and agricultural industries worldwide. Understanding the behaviour and ecology of these beetles is crucial for developing effective management strategies to mitigate their impacts [20].

The rubber tree, *Hevea brasiliensis* Müll. Arg. (Euphorbiaceae), is a species indigenous to the Amazon basin, now widely introduced across tropical regions due to its rapid growth and high latex yield [21]. Sabah, a Malaysian state situated at the northern tip of Borneo Island, initiated rubber cultivation in 1882, with significant expansion occurring after 1905 as its economic viability was recognized [22]. Currently, the rubber tree is among the most extensively planted tree species in Sabah [23], primarily for latex extraction and timber production. The wood from rubber trees is commonly utilized in making furniture [24]. Additionally, rubber plantations are vital to the rural economy, providing substantial income and enhancing the living standards of local communities [25,26].

Bark and ambrosia beetles (Coleoptera: Curculionidae: Scolytinae and Platypodinae), along with powder post beetles (Coleoptera: Bostrichidae), are recognized as significant pests of rubberwood sawn timber [27]. Bark and ambrosia beetles thrive in moist conditions, while powder post beetles exhibit a preference for dry wood [28]. The infestation by these beetles can lead to a decline in forest health and significantly deteriorate the quality of rubberwood, ultimately reducing its commercial viability. Several studies have documented species of bark and ambrosia beetles infesting rubberwood timber [27,29]. For instance, Sittichaya and Beaver [27] identified 21 species of wood-boring beetles infesting rubberwood, including 10 from the family Bostrichidae, nine from the subfamily Scolytinae, and two from the subfamily Platypodinae. Additionally, an unidentified species from the genus *Euplatypus* belonging to the subfamily Platypodinae was reported to cause stem bleeding in rubber trees in the Philippines [30].

Despite their economic importance, the knowledge of bark and ambrosia beetle species that attack rubber trees in Sabah remains elusive. Chung et al. [23] documented pests, including three scolytine species, in various tree plantations in Sabah, including rubber plantations. However, there was no information indicating whether these three species specifically target rubber trees. This knowledge gap is particularly concerning given Sabah’s rich biodiversity and the difficulty in taxonomy of these groups due to their small body size and cryptic behaviour.

To address this issue, a comprehensive long-term study was conducted using ethanol-baited traps over approximately three years in three distinct types of forest: primary forest, disturbed forest and rubber plantations within a tropical rainforest in southern Sabah, Malaysia. The sample coverage and the percentage of recorded species were estimated. Several diversity indices of Scolytinae and Platypodinae were calculated. Moreover, we performed indicator species analysis to identify specific beetle species that were strongly associated with each forest type, providing insights into habitat preferences and potential ecological indicators. The beetle assemblage was compared among the three forest types to determine anthropogenic influence on the Scolytinae and Platypodinae assemblage.

## 2. Materials and Methods

### 2.1. Data Sources

The data used in this study were obtained in southern Sabah, Malaysia (4°23′–27′ N, 115°42′–47′ E) and were published as a data paper [31]. Bark and ambrosia beetles were sampled using traps baited with ethanol, which were modified from Steininger et al. [32] and exactly the same as used by Sanguansub et al. [33]. The trap consisted of a 500-mL plastic water bottle, a 10-mL conical tube, and a plastic tray (Figure 1). A rectangular opening measuring approximately 5 cm × 10 cm was cut into the bottle to allow beetles access. Ethanol (95% *v*/*v*) was placed inside the conical tube as an attractant. Propylene glycol was added to the bottle to quickly kill the beetles and preserve their condition. A plastic tray was positioned at the top of the trap to serve as a roof. Four ethanol-baited traps were set at each of the three forest types: primary forest (PF), disturbed forest (DF), and rubber plantation (RP; PL in Gunggot et al. [31]) for three years from 29 April 2017 to 30 May 2020 (Table 1). The distance between PF and RP was approximately 600–700 m, while DF was situated about 2–3 km away from both PF and RP. The rubber trees in the RP were planted 2–3 years before starting the beetle trapping. One hundred thirty-four species belonging to the subfamily Scolytinae (7232 individuals) and 20 species belonging to the subfamily Platypodinae (25 individuals) were recorded in 80 biweekly samples [31] and analysed in this study. Further details of the methodology, including study sites and trap designs, can be found in the data paper [31].

### 2.2. Statistical Analysis

All the data analyses were performed using R software (ver. 4.4.0) [34].

Rarefaction and extrapolation analyses were conducted to estimate sample coverage based on Chao-1 estimator, an abundance-based metric, of 80 biweekly samples and up to double the observed sample size. Species richness was also estimated based on Chao-1 estimator. The Chao-1 estimator was selected due to the prevalence of rare species (singletons, doubletons, tripletons, …) [35], as it is particularly effective when rare species are abundant. The percentages of species observed in the 80 samples, and with up to double the observed sample sizes (160 times of samples), were estimated. These results were compared across PF, DF, and RP. These analyses were performed using “iNEXT package (ver. 3.0.1)” [36]

To assess and compare species diversity across forest types, the Shannon-Wiener diversity index (*H*′) and the Berger-Parker dominance index (*D*) along with abundance and species richness were determined for each forest type and the overall of three forest types together. To evaluate dissimilarity in species composition among PF, DF, and RP, the Bray-Curtis dissimilarity index and Chao dissimilarity index were employed. The Chao index accounted for rare, unseen shared species, making it suitable for communities with abundant rare species [37], while the Bray-Curtis index measured dissimilarity based on relative abundances. All analyses were performed using the package “vegan (ver. 2.6-6.1)” [38]. To test the difference in these diversity indices among the sites, a bootstrap resampling approach was employed to generate 10,000 replicates of 80 bi-weekly samples from the dataset. The abundance, species richness, *H*′ and *D* values, and two dissimilarity index values were then calculated for each sample. Mean values and 95% confidence intervals were estimated from mean and standard deviation of the 10,000 values. Tukey’s HSD test was utilized for multiple comparisons following the analysis of variance. The package “dplyr (ver. 1.1.4)” [39] and the package “vegan (ver. 2.6-6.1)” [38] were used for resampling and calculating the index values, respectively. Rank-abundance curves were drawn for each forest type. The numbers of species that were collected > 10, >100, and >1000 individuals, along with singleton (species with exactly one individual), doubleton (species with exactly two individuals), and tripleton species (species with exactly three individuals) in each forest type were determined.

To quantify and illustrate shared and unshared species across PF, DF, and RP, a Venn diagram was generated using the package “VennDiagram (ver. 1.7.3)” [40]. The numbers of singletons, doubletons, and tripletons that were captured only from one forest type were determined.

Indicator species analysis was conducted to identify indicator species for each of the forest types (PF, DF, and RP). The selection criteria, as outlined by Dufrene and Legendre [41], stipulated that only species with an indicator value exceeding 25% (*p* < 0.05) were considered as indicator species. The analysis utilized the package “Indicspecies (version 1.7.14)” [42] to compute indicator values, while the package “permute (ver. 0.9-7)” [43] was used to perform permutation tests for statistical validation.

## 3. Results

### 3.1. Bark and Ambrosia Beetles Species Composition

The rank abundance curve declined most steeply in PF followed by DF (Figure 2). In the PF, *Eidophelus* (*Scolytogenes*) sp. SA1 emerged as the most abundant species, representing 51.5% of the total captures with 1389 individuals. This was followed by *Dryocoetiops moestus* (Blandford) (274 individuals; 10.2%) and *Eidophelus* sp. SA1: (a distinct species from *Eidophelus* (*Scolytogenes*) sp. SA1) (111 individuals; 4.1%). One species had over 1000 individuals, three species had more than 100 individuals, and 21 species had more than 10 individuals. Additionally, 41 species had more than 3 individuals. Additionally, the PF contained 10 tripletons, 12 doubletons, and 33 singletons.

In the DF, the first and second dominant species were the same as the PF although the dominance by *Eidophelus* (*Scolytogenes*) sp. SA1 (521 individuals; 35.4%) was lower than the PF. On the contrary, the relative abundance of the second dominant species (*D. moestus*:196 individuals; 13.4%) was greater. The third dominant species was *Ambrosiodmus asperatus* (Blandford) (170 individuals; 11.6%). The DF had no species with over 1000 individuals, but three species had more than 100 individuals. It also featured 16 species with more than 10 individuals and 30 species with more than three individuals. The DF contained 6 tripletons, 15 doubletons, and 41 singletons. The number of singletons was greatest in the DF.

In the RP, the dominant species, *Eidophelus* (*Scolytogenes*) sp. SA1 was least dominant overall (834 individuals; 27.0%). The second dominant species in RP was *Hypothenemus eruditus* cx Westwood (619 individuals; 20.0%) with the greatest relative abundance among the second dominant species in the three forest types. The third dominant species was *Eccoptopterus spinosus* (Olivier) (241 individuals; 7.8%). The RP also had no species exceeding 1000 individuals but eight species surpassing 100 individuals, the highest among the three forest types. RP also comprised 25 species with a population of above 10 individuals and 41 species with more than three individuals. In terms of rareness, RP contained 3 tripletons, 14 doubletons, and 39 singletons. In RP, only the most dominant species, *Eidophelus* (*Scolytogenes*) sp. SA1 was the same with PF and DF. *H. eruditus* cx, the second most abundant species in RP (619 individuals), was the 6th (64 individuals) and the 9th (35 individuals) in PF and DF, respectively. Similarly, *E. spinosus*, the third most abundant species (241 individuals), was the 6th (64 individuals) and the 4th (66 individuals) in PF and DF, respectively. *Hypothenemus areccae* cx (Hornung), the fourth most abundant species (229 individuals), was the 9th (36 individuals) and the 8th (39 individuals) in PF and DF, respectively. The species that were recorded from rubber trees (see Discussion) increased greatly in the RP in abundance as well as in their relative ranks. A similar tendency was found in *Hypothenemus birmanus* (Eichhoff), the ninth species in RP (68 individuals), but only four and two individuals in PF and DF, respectively.

### 3.2. Trap Captures, Diversity Indices, and Species Estimation

The number of individuals and species richness were greatest in RP followed by PF (Table 2). The number of individuals in RP was more than twice higher than DF, while the variation in species richness was minimal. The Shannon-Wiener diversity index (*H*′) was greatest in DF closely followed by RP. The Berger-Parker dominance index (*D*) was greatest in PF followed by DF. The *D* value in PF was greater than 50%. Bootstrap values of all the four indices were significantly different between the three sites (Tukey’s HSD test, *p* < 0.001). The bootstrap values of the four diversity indices among the three forest types (PF, DF, and RP) exhibited a pattern similar to the observed values. Although some bootstrap values of *H*′ may exceed the observed *H*′, the number of species—a component of the *H*′ formula—limited the mean of the bootstrap values. Specifically, since the bootstrap values for species richness do not exceed the observed species richness, there is a smaller mean of the bootstrap values of *H*′ compared to the observed *H*′.

The rarefaction curves indicated that species richness was saturated at very early stages of sampling (approximately 150 individuals) compared to the total number sampled (<10%), after which the rate of increase slowed considerably (Figure 3). By the 80th sampling period, sample coverage was 98.8%, 97.2%, and 98.7% for PF, DF, and RP, respectively. The extrapolation curves exhibited milder slopes compared to the rarefaction curves. The estimated sample coverages reached 99.4%, 98.7%, and 99.4% for PF, DF, and RP, respectively, when considering a sample size that was doubled through an additional 80 sampling events.

Across the three study sites (PF, DF, and RP), species richness estimated by Chao1-estimator exhibited substantially greater values (141–152) than observed values (94–98), with sampling completeness ranging from 64.8% to 70.6% of the estimated number of species (Table 3). Despite high sample coverage (≈98%), 45–55 unobserved species were detected at each site over the three-year sampling period.

### 3.3. Dissimilarity of Bark and Ambrosia Beetles in Three Types of Forest

Although PF and RP were geographically closer, both Bray-Curtis and Chao dissimilarity were not smallest between these forests (Table 4). The Bray-Curtis dissimilarity index was the highest between DF and RP, followed closely by PF and RP. The lowest dissimilarity occurred between PF and DF, indicating that RP had the most distinct community structure. Significant differences were identified in all combinations of the pairs of the three bootstrap values of the Bray-Curtis index (Tukey’s HSD test, *p* < 0.001). In contrast, the Chao dissimilarity index revealed a different pattern, showing the highest dissimilarity between PF and DF, followed closely by PF and RP, while the lowest dissimilarity was noted between DF and RP. Bootstrap values of Chao dissimilarity index exhibited significant differences across all combinations of the pairs (Tukey’s HSD test, *p* < 0.001). Notably, Chao dissimilarity index values were much smaller than those of Bray-Curtis suggesting great proportion of unobserved species. Small values of the Chao dissimilarity index (<0.04) indicate that the three forest types harbored similar fauna of Scolytinae and Platypodinae when considering the unseen species.

Among the 154 species, there were 49 species shared among the three forest types, with 21 species unique to DF, 22 species to PF, and 26 species to RP (Figure 4). Additionally, 11 species were shared between DF and RP, 12 species between RP and PF, and 13 species between DF and PF. Regarding the species unique to each of the three forest types, most of these were singletons (Table 5). Two doubletons (*Arixyleborus* sp. 2 and *Ptilopodius* sp. SB01) and two tripletons (*Ancipitis puer* (Eggers) and *Platypus pasaniae* Schedl) were recorded only in PF. Two doubletons (*Cryptoxyleborus cuneatus* Beaver & Hulcr and *Truncaudum agnatum* (Eggers)) and one tripleton were recorded only in DF. Six doubletons were recorded only in RP. These included two unidentified species belonging to the genus *Cryphalus* (SB04 and SB06), *Ernocladius* sp. SB01, *Ozopemon brownei* Schedl, *Eidophelus* (*Scolytogenes)* sp. SB02, and *Sueus borneensis* Bright. Species captured in numbers greater than three individuals were shared among two or more types of forests.

Three species were identified as indicator species for PF and six species for RP, whereas no indicator species were detected for DF (Table 6). Regarding the indicator species for RP, four were species belonging to the genus *Hypothenemus* including *H. eruditus* cx, and two to the genus *Eccoptopterus*.

## 4. Discussion

### 4.1. High Diversity and Stochasticity of Bark and Ambrosia Beetles in Tropical Rainforest

Tropical rainforests are among the most biodiverse ecosystems on Earth [1]. Their complex structure, stable environment, and intricate interdependencies between flora and fauna contribute to their extraordinary species richness [44]. The diversity of insects in Sabah is particularly remarkable, with numerous species adapted to various ecological niches [45]. The number of scolytine and platypodine species captured in this study was nearly twice as high as those found in tropical seasonal forests in Thailand using the same trap design [33]. In that study, seven species of Platypodinae and 69 species of Scolytinae were collected from 77 biweekly samplings employing 12 traps. In contrast, the number of species captured by EtOH-baited traps is significantly smaller in the temperate zone: 17 scolytine species were collected over one year in Ohio and Virginia, USA, with 16 collected in the subsequent year in Ohio [46]. Additionally, 20 scolytine species and one platypodine species were recorded over two years in Kyoto, Japan [47]. This indicates that the fauna of Scolytinae and Platypodinae in the tropical rainforest of Sabah is highly diverse compared to other forest ecosystems.

Despite high sampling coverage (≈98%), rarefaction curves and the Chao-1 estimator suggested the presence of many unobserved species (≤70.6% of the estimated total; Table 3). This implies that the beetle community exhibits high stochasticity, likely due to the prevalence of rare species. The contrasting results of the Bray-Curtis and Chao dissimilarity indices further support this notion; while the Bray-Curtis index highlighted differences in abundant species, the Chao index indicated greater homogeneity across the three forest types because the Chao index considered the stochasticity [48,49]. These findings underscore the limitations of current sampling methods and emphasize the need for sustained efforts to fully uncover the extent of biodiversity.

### 4.2. Diversity Indices Among Three Forest Types

Most bark and ambrosia beetles are secondary or saprophytic insects that primarily target unhealthy, weakened, or dead trees [13], making resource availability a key determinant of their abundance. The lower abundance and species richness observed in disturbed forests (DF) compared to primary forests (PF) likely reflect reduced resource availability, exacerbated by human activities such as the removal of deadwood for fuel. Additionally, the smaller tree size in DF compared to PF [50], due to selective logging, likely contributed to this outcome.

Conversely, rubber plantations (RP) exhibited the highest abundance and species richness despite their monocultural nature and the lowest biomass. This counterintuitive result can be attributed to RP’s open canopy structure, which likely enhances beetle flight activity, as noted in studies on forest gaps and roads [51,52]. Moreover, RP’s proximity to PF may have facilitated beetle migration, while the presence of rubber tree-specific species (discussed in Section 4.3) further contributed to the observed patterns.

*Eidophelus* (*Scolytogenes*) sp. SA1 was the most dominant species across the three forest types. Following taxonomic revisions, *Eidophelus* was reclassified, including the synonymization of *Scolytogenes* within *Eidophelus* [53]. Some species previously classified under the genus *Scolytogenes* are known to inhabit the petioles of large leaves as well as the area beneath the bark [54]. This characteristic is a likely cause of the highest dominance of *Eidophelus* (*Scolytogenes*) sp. SA1 in PF among the three forest types, which may be linked to the greater availability of petioles in that area, as the trees are larger. This increased resource availability may facilitate the higher capture rates of this species compared to other sites; however, this remains speculative and warrants further investigation.

The Shannon-Wiener diversity index (*H*′) was highest in DF, followed closely by RP. The elevated *H*′ value in RP can be explained by its lowest *D* value and the highest species richness, with more than 100 individuals captured, the most among the three sites. In contrast, the highest number of singletons in DF may have been a key factor contributing to its elevated *H*′ value.

### 4.3. Influence of Rubber Trees on Bark and Ambrosia Beetle Assemblage

Community similarity did not correspond to geographical proximity, as evidenced by the Bray-Curtis and Chao dissimilarity indices. Forest type strongly influenced beetle assemblages, with rubber plantations (RP), in particular, exhibiting a distinct community composition. Notable species, such as *H. eruditus* cx and *E. spinosus*, dominated the RP.

The high abundance of *H. eruditus* cx can be attributed to its remarkable adaptability, as it inhabits diverse substrates, including leaf petioles, twigs, other beetle galleries, fungal fruiting bodies, and even manufactured objects [55,56,57]. However, this does not explain why *H. eruditus* was less abundant in PF and DF. The rubber tree is a non-native species introduced to Malaysia in the 18th century [22,23]. The RP was established by converting a portion of PF approximately 2 to 3 years prior to the commencement of this study. Consequently, rubber trees were found exclusively in RP among the study sites. The presence of this non-native species in RP likely contributed to the unique species composition observed at this site.

Sittichaya and Beaver [28] and Kangkamanee et al. [29] reported that beetles bored into the wood of felled rubber trees in Thailand, identifying five species in the subfamily Platypodinae and nine species in the subfamily Scolytinae. Among these, eight species were recorded in the rubber plantation (RP). Notably, the top four of these eight species, ranking 2nd to 9th in dominance within the RP, exhibited significantly greater abundance and higher relative ranks in the RP compared to the other two forest types. This finding strongly suggests that the presence of rubber trees likely promoted the increased abundance of beetle species capable of colonizing them.

Among the eight species, four species (*H. eruditus* cx, *E. spinosus*, *H. areccae* cx, and *H. birmanus*) were identified as significant indicators of the RP, indicating that their populations may have benefited from the availability of rubber trees. Interestingly, two additional indicator species of the RP, *Hypothenemus* sp. SB06 and *Eccoptopterus limbus* Sampson, had not been previously documented in studies of rubber tree-associated beetles. Although the mechanisms underlying the identification of these two species as RP indicators remain unclear, it is plausible that they infest rubber trees. Notably, relatively few species of bark and ambrosia beetles are known to attack healthy trees [13], suggesting that the rubber trees in Long Miau were likely less vigorous, thereby attracting bark and ambrosia beetles capable of infesting stressed or weakened trees.

## 5. Conclusions

The Scolytinae and Platypodinae assemblages captured by ethanol-baited traps revealed anthropogenic influences on beetle communities in tropical forest ecosystems. Higher diversity was observed in DF and RP compared to PF probably due to anthropogenic effects. The species composition in RP was most distinct among the three forest types, likely due to the introduction of non-native tree species. The indicator species of RP included both known rubber tree-infesting species and species not previously reported from rubber trees. The findings demonstrate the human impact of land-use change and introduction of non-native host species on insect community structure.

## Figures and Tables

**Figure 1 insects-16-00121-f001:**
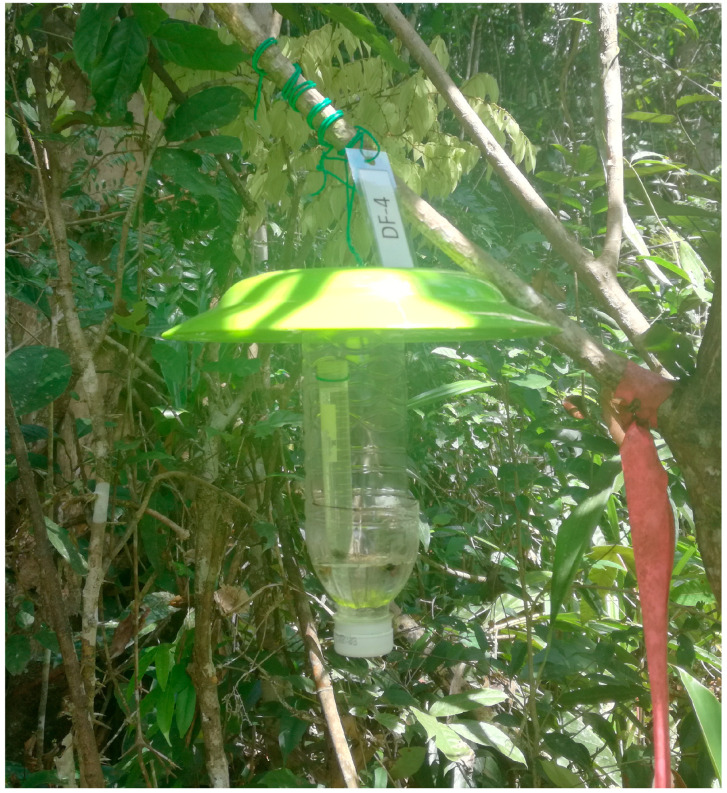
An ethanol-baited trap hung on a tree branch to capture bark and ambrosia beetles in Long Miau, Sipitang, Sabah, Malaysia. A 10-mL conical tube filled with 95% ethanol was placed inside the trap. Approximately 50 mL of propylene glycol was added to the bottom of the trap to kill insects rapidly and preserve their condition. A plastic tray was positioned at the top of the trap to serve as a roof.

**Figure 2 insects-16-00121-f002:**
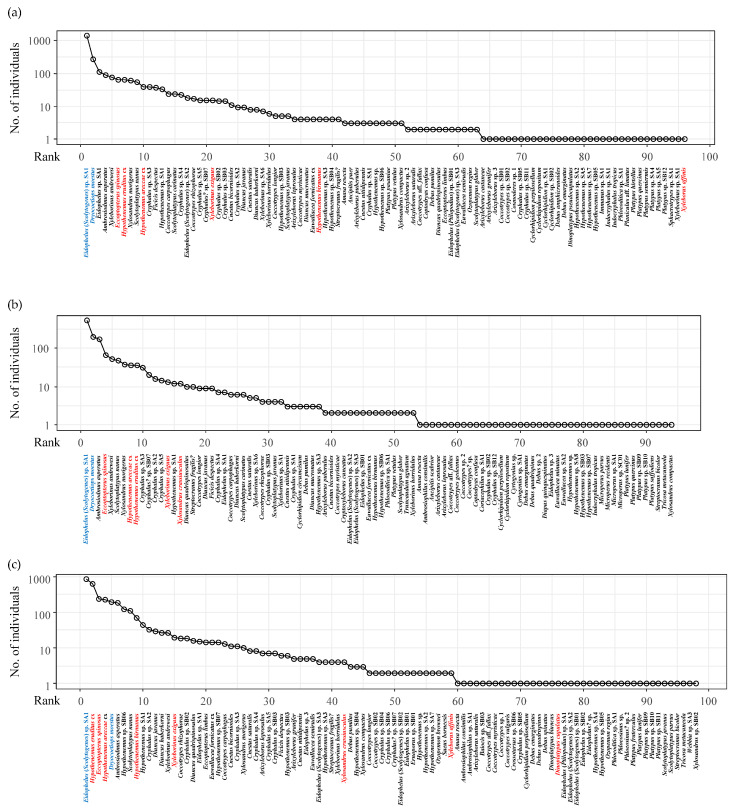
Rank abundance curves of the subfamilies Scolytinae and Platypodinae captured by the four ethanol-baited traps installed at three different types of forests (PF: Primary Forest (**a**), DF: Disturbed Forest (**b**), and RP: Rubber Plantation (**c**)) in Long Miau, Sabah, Malaysia from April 2017 to May 2020. Blue letters indicate the first and second most dominant species in both PF and DF, while red letters highlight species previously recorded from rubber trees in other studies.

**Figure 3 insects-16-00121-f003:**
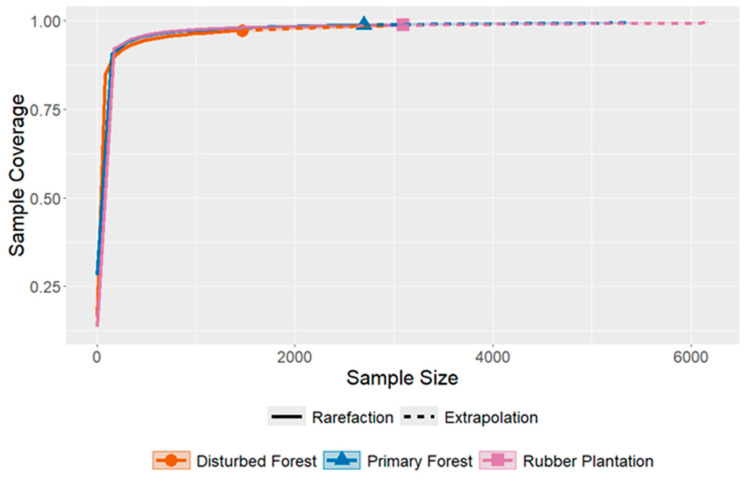
Rarefaction and extrapolation curves for the number of species of the subfamilies Scolytine and Platypodine based on individual-based abundance data showing sample completeness by 80 times of biweekly sampling at three different types of forest in Long Miau, Sabah, Malaysia from April 2017 to May 2020.

**Figure 4 insects-16-00121-f004:**
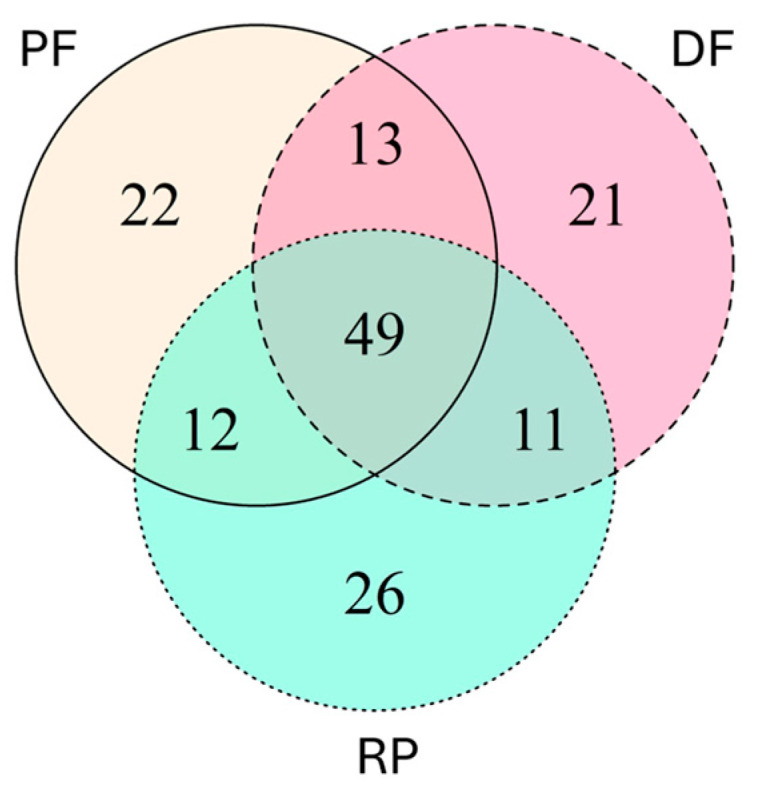
Venn-diagram of the number of Scolytinae and Platypodinae species captured by the four ethanol-baited traps installed at three different types of forest (PF: Primary Forest, DF: Disturbed Forest, and RP: Rubber Plantation) in Long Miau, Sabah, Malaysia from April 2017 to May 2020.

**Table 1 insects-16-00121-t001:** Elevation and coordinate for 12 ethanol-baited traps deployed across three forest types in Long Miau, Sipitang, Sabah, Malaysia from 29 April 2017 to 30 May 2020 (after Gunggot et al. [31]).

Type of Forest	Trap	Elevation (m a.s.l.)	Coordinates
Primary Forest (PF)	PF−1	1082	4°26′52.6″ N 115°44′11.4″ E
	PF−2	1078	4°26′51.7″ N 115°44′11.5″ E
	PF−3	1098	4°26′50.4″ N 115°44′09.5″ E
	PF−4	1104	4°26′50.4″ N 115°44′08.4″ E
Disturbed Forest (DF)	DF−1	1024	4°27′27.0″ N 115°44′50.0″ E
	DF−2	1016	4°27′23.7″ N 115°44′50.0″ E
	DF−3	1065	4°27′41.5″ N 115°44′34.0″ E
	DF−4	1064	4°27′41.2″ N 115°44′34.1″ E
Rubber Plantation Forest (RP) *	RP−1	1081	4°26′51.7″ N 115°44′15.5″ E
	RP−2	1081	4°26′51.4″ N 115°44′16.0″ E
	RP−3	1047	4°26′48.4″ N 115°44′22.5″ E
	RP−4	1047	4°26′47.9″ N 115°44′23.0″ E

*: Plantation Forest (PL) in Gunggot et al. [31].

**Table 2 insects-16-00121-t002:** Abundance, species richness, Shannon-Wiener diversity index, and Berger-Parker dominance index with bootstrap value with 95% confidence intervals (indicated in parentheses) for the subfamilies Scolytinae and Platypodinae captured by four ethanol-baited traps installed at three different types of forest (PF: Primary Forest, DF: Disturbed Forest, and RP: Rubber Plantation) in Long Miau, Sabah, Malaysia from April 2017 to May 2020. The same letter indicates no difference between the groups. Details and illustrations of bootstrap values comparisons are included in Appendix A.

	PF	DF	RP	Overall
Number of individuals	2697 (2698.9 ± 475.8) ^a^	1468 (1469.0 ± 253.0) ^b^	3092 (3093.8 ± 400.9) ^c^	7257
Species richness	96 (79.7 ± 9.3) ^a^	94 (75.2 ± 8.0) ^b^	98 (80.1 ± 7.7) ^c^	154
Shannon-Wiener diversity index (*H*′)	2.294 (2.27 ± 0.15) ^a^	2.675 (2.626 ± 0.12) ^b^	2.637 (2.610 ± 0.08) ^c^	2.681
Berger-Parker dominance index (*D*)	0.515 (0.514 ± 0.040) ^a^	0.354 (0.353 ± 0.034) ^b^	0.269 (0.270 ± 0.044) ^c^	0.377

**Table 3 insects-16-00121-t003:** Number of species of the subfamilies Scolytinae and Platypodinae estimated by Chao-1 species estimator for three different types of forest (PF: Primary Forest, DF: Disturbed Forest, and RP: Rubber Plantation) in Long Miau, Sabah, Malaysia using data obtained from 80 biweekly samples from April 2017 to May 2020. The variation indicates a standard deviation of each estimate.

Species	PF	DF	RP
Observed	96	94	98
Estimated	141 ± 21.6	149 ± 23.9	152 ± 23.8
% Observed	70.6	64.8	66.7

**Table 4 insects-16-00121-t004:** Bray-Curtis and Chao Dissimilarity indices with bootstrap value (indicated in parentheses) among three different types of forest for Scolytinae and Platypodinae assemblages captured by the four ethanol-baited traps installed at three different types of forest (PF: Primary Forest, DF: Disturbed Forest, and RP: Rubber Plantation) in Long Miau, Sabah, Malaysia from April 2017 to May 2020. The same letter indicates no difference between the groups. Details and illustrations of bootstrap values comparisons are included in Appendix A.

	PF–DF	PF–RP	DF–RP
Bray-Curtis Dissimilarity Index	0.375 (0.394 ± 0.067) ^a^	0.435 (0.444 ± 0.072) ^b^	0.439 (0.454 ± 0.059) ^c^
Chao Dissimilarity Index	0.035 (0.0492 ± 0.033) ^a^	0.033 (0.0557 ± 0.049) ^b^	0.016 (0.071 ± 0.056) ^c^

**Table 5 insects-16-00121-t005:** Number of singletons, doubleton, and tripleton species belonging to the subfamilies Scolytinae and Platypodinae unique to each of the three forest types, which were captured by the four ethanol-baited traps installed at three different types of forest in Long Miau, Sabah, Malaysia from April 2017 to May 2020.

	PF	DF	RP
Singletons	18	18	20
Doubletons	2	2	6
Tripletons	2	1	0
Total	22	21	26

**Table 6 insects-16-00121-t006:** Indicator species belonging to the subfamilies Scolytinae and Platypodinae for each of the three forest types (PF: Primary Forest, DF: Disturbed Forest, and RP: Rubber Plantation) in Long Miau, Sabah, Malaysia identified by the indicator species analysis. According to Dufrêne and Legendre [41], species with an indicator value greater than 25% (*p* < 0.05) were identified as indicator species.

Forest Type	Species	Indicator Value	*p*-Value
PF	*Eidophelus* (*Scolytogenes*) sp. SA1	0.319	0.0001
	*Scolytoplatypus carinatus* Bright	0.306	0.0001
	*Ficicis despectus* (Walker)	0.268	0.0001
RP	*Hypothenemus eruditus* cx Westwood	0.582	0.0001
	*Hypothenemus areccae* cx (Hornung)	0.468	0.0001
	*Hypothenemus* sp. SB06	0.412	0.0001
	*Hypothenemus birmanus* (Eichhoff)	0.397	0.0001
	*Eccoptopterus limbus* Sampson	0.273	0.0003
	*Eccoptopterus spinosus* (Olivier)	0.265	0.0001

## Data Availability

Insect specimens used for this publication are available at laboratory of Entomology, Faculty of Tropical Forestry, Universiti Malaysia Sabah. Original dataset has been published in the following publication. Evahtira Gunggot, Roger A Beaver, Jonathan Jimmey Lucas, Sandra Georgina George, Anastasia Rasiah, Wilson VC Wong, Maria Lourdes T Lardizabal, Naoto Kamata. List of Scolytinae and Platypodinae (COLEOPTERA: Curculionidae) captured by ethanol-baited traps in a tropical rainforest in southern Sabah, Malaysia. Miscellaneous Information of the University of Tokyo Forests, 2025, 71, in press.

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
