# Peer review of "Anthropogenic Impacts on Bark and Ambrosia Beetle Assemblages in Tropical Montane Forest in Northern Borneo"

_insects, 2025, doi:10.3390/insects16020121_

Round 1

Reviewer 1 Report

Comments and Suggestions for Authors

This study examined bark and ambrosia beetles in three forest types in southern Sabah, Malaysia, using ethanol-baited traps. The study gives information about the role of human intervention affecting the diversity and distribution of wood boring beetles. Data were collected for three years, which shows the robustness of this work. While the study has been neatly executed, the authors need to spend some more time to improve the presentation.

Comments:

Title: The title is too long for a paper. I would suggest revising it and making it concise. For example:

Diversity and Distribution of Bark and Ambrosia Beetle Assemblages across Primary, Disturbed, and Rubber Forests in Ulu Padas, Malaysia.

Simple summary: As mentioned in the template of MDPI insects, you need to come up with concise and simple summary. Please make sure that you are writing it for an audience which does not have any previous background in this subject. Read the following and I would suggest you visit the example given in the link. For your ease, I am mentioning the guidelines in the following lines:

It is vitally important that scientists are able to describe their work simply and concisely to the public, especially in an open-access on-line journal. The simple summary consists of no more than 200 words in one paragraph and contains a clear statement of the problem addressed, the aims and objectives, pertinent results, conclusions from the study and how they will be valuable to society. This should be written for a lay audience, i.e., no technical terms without explanations. No references are cited and no abbreviations. Submissions without a simple summary will be returned directly. Example could be found at https://www.mdpi.com/2075-4450/11/8/508f (Reference: MDPI Insects Template).

L26: When you start the abstract, you start with a background statement. In this case, you have not mentioned it. Add a statement which links to the anthropogenic disturbances and changes in the species distribution.

L26-45: The abstract is too long. As per Insects MDPI format, it should be around 200 words. Revise the abstract and make sure that it has the standard sections, i.e., background/introduction, methods, results, conclusion including recommendations or future directions, if possible. See the following description:

A single paragraph of about 200 words maximum. For research articles, abstracts should give a pertinent overview of the work. We strongly encourage authors to use the following style of structured abstracts, but without headings: (1) Background: Place the question addressed in a broad context and highlight the purpose of the study; (2) Methods: briefly describe the main methods or treatments applied; (3) Results: summarize the article’s main findings; (4) Conclusions: indicate the main conclusions or interpretations. The abstract should be an objective representation of the article, and it must not contain results that are not presented and substantiated in the main text and should not exaggerate the main conclusions (Reference: MDPI Insects Template).

L63-64: Revise the starting of the second paragraph. Be more specific as it is unclear what are the economic and ecological impacts. Are these impacts positive or negative? What exactly are you referring to?

L79: When you are writing a species for the first time, write in the following format: Ambrosia beetle, Euwallacea fornicatus (Eichhoff) (Coleoptera: Curculionidae). Later on, you will just use E, fornicates. However, in figures and tables and their captions, you will use the complete format.

L87: When you write the insect species for the first time, mention family in brackets after the common name, scientific name, authority. For example: Rubber tree, Hevea brasiliensis Müll.Arg. (Euphorbiaceae). Later on, you will just use, H. brasiliensis. However, in figures and tables and their captions, you will use the complete format.

L96: Also write the family and order of bark and ambrosia beetles.

L115-117: Revise this sentence. Mention clearly the three types of study areas.

L119-121: Replace: Moreover, we performed indicator species analysis to identify specific beetle species that were strongly associated with each forest type, providing insights into habitat preferences and potential ecological indicators.

Materials and methods

This is a three-year study, and I am amazed to see that the authors have completely ignored the materials and methods portion. A new researcher will not be able to replicate this study in any case. This will decrease the future citations of your paper. I would suggest the following papers and revise your methodology accordingly. I would strongly recommend citing both papers.

Monterrosa, A., Joseph, S. V., Blaauw, B., Hudson, W., & Acebes-Doria, A. L. (2022). Ambrosia beetle occurrence and phenology of Xylosandrus spp.(Coleoptera: Curculionidae: Scolytinae) in ornamental nurseries, tree fruit, and pecan orchards in Georgia. Environmental Entomology51(5), 998-1009.

https://academic.oup.com/ee/article/51/5/998/6674462

Govindaraju, R., Hayter, J., Chong, J. H., Del PozoValdivia, A. I., Cottrell, T. E., Walgenbach, J. F., ... & Joseph, S. V. (2024). Influence of the Ethanol Lure and Concentration on Captures of Ambrosia Beetles in Tree Fruits and Ornamentals. Journal of Applied Entomology.

https://onlinelibrary.wiley.com/doi/full/10.1111/jen.13361

Also, create a table summarizing the trapping locations, trapping periods, and forest types. This will help the reader easily identify each site.

Add pictures of traps, as given in the above-mentioned papers.

L135-175: Statistical analysis section of materials and methods looks good.

L374 and L376: Write correct spellings for individuals.

Discussion: I have read the discussion, and it feels like I am reading the results and discussion together. I suggest a complete revision of this section and focus on the top results for discussion. This section is not to rewrite your results rather you have to discuss why the results came up in a certain way. Simply, just follow the standard discussion protocol. Remember, there is a difference between a dissertation and a research paper style. The discussion (which looks like part of a dissertation) already has many exciting points, but it lacks a standard format. See the above mentioned papers to improve the discussion part.

Author Response

Dear Reviewer,

Thank you for your valuable suggestions and comments. We have revised the manuscript accordingly. Please find attached the revised file along with our detailed responses.

You requested a revision of the M&M section. However, the detailed methodology is described in a data paper that was accepted on November 20, 2024, and will be published in 2025. Therefore, we provide a concise description of the M&M section, excluding statistical analyses. The accepted draft has also been uploaded for reference.

We appreciate your time and effort in reviewing our work.

Best regards,
Naoto Kamata

Reviewer 2 Report

Comments and Suggestions for Authors

A very interesting paper describing the impact of human management (in this case rubber tree plantations) on the Scolytinae and Platypodinae assemblages. For this purpose, the authors compared rubber tree plantations to forests disturbed by humans and primeval forests. The paper is of great importance in the conditions of rapid reduction of rainforest areas throughout the equatorial zone.

My comments are more technical than substantive in nature and are included in the text of the work.

Author Response

Dear Reviewer,

Thank you for your valuable suggestions and comments. We have revised the manuscript accordingly. Please find attached the revised file along with our detailed responses.

We appreciate your time and effort in reviewing our work.

Best regards,
Naoto Kamata

Round 2

Reviewer 1 Report

Comments and Suggestions for Authors

I encourage the authors to write the complete materials and methods using a sub-heading method. It should be detailed and complete.

Author Response

TO: Reviewer 1,

I would like to express my sincere gratitude for your valuable feedback on our manuscript. We have carefully considered your suggestions and have incorporated them where feasible. However, we respectfully disagree with the recommendation to undertake a complete rewrite of the detailed methodology, as this may lead to concerns regarding self-plagiarism. To address this issue, we sought the opinion of an academic editor, who also recommended the inclusion of a table summarizing the trapping locations, trapping periods, and forest types, as suggested in your initial review. We have added this information in the form of a new Table 1 within the Materials and Methods section.

We believe that this approach adequately addresses your request while preserving the integrity of our original work.

Furthermore, after an additional review of the manuscript, we have decided to revise the title once more. We believe that the new title is more closely aligned with the overall flow and content of the paper. We apologize for any oversight in our previous review of the manuscript.

Thank you once again for your time and consideration. We hope that these revisions meet your expectations and are in accordance with the journal's standards.

Sincerely,

Naoto Kamata

Colour Code

Round 3

Reviewer 1 Report

Comments and Suggestions for Authors

The authors have significantly revised the manuscript.